# Current status of cervical cytology during pregnancy in Japan

**Shunji Suzuki**[1,2]*, **Eijiro Hayata**[2], **Shin-ichi Hoshi**[2], **Akihiko Sekizawa**[2], **Yoko Sagara**[2], **Masanobu Tanaka**[2], **Katsuyuki Kinoshita**[2], **Tadaichi Kitamura**[3]

**1** Department of Obstetrics and Gynecology, Japanese Red Cross Katsushika Maternity Hospital, Tokyo, Japan, **2** Japan Association of Obstetricians and Gynecologists, Tokyo, Japan, **3** Japanese Foundation for Sexual Health Medicine, Tokyo, Japan

* czg83542@mopera.ne.jp

**Data Availability Statement:** The data are published in figshare (10.6084/m9.figshare. 13347299, Title: Current status of cervical cytology in Japan).

## Abstract

In Japan, uterine cancer screening during pregnancy is subsidized by public funds. We examined the current status of the results of cervical cytology conducted during pregnancy in Japan. We requested 2,293 obstetrical facilities to provide information on cervical cytology in pregnant women who delivered between October 2018 and March 2019. A total of 1,292 obstetrical facilities responded, with valid information on a total of 238,743 women. The implementation rate of cervical cytology during pregnancy was 86.8% in Japan. The prevalence of abnormal cervical cytology during pregnancy was 3.3% in total and 4.9% using a spatula/brush with liquid-based cytology (LBC). The prevalence of positive high-risk human papillomavirus (HPV) in teenagers with atypical squamous cells of undetermined significance (ASC-US) was significantly higher than women of other ages ($p < 0.01$). Because HPV vaccine coverage has dropped to less than 1% in Japan, a further study with various conditions will be needed to improve the accuracy of cervical cancer screening during pregnancy.

## Introduction

Uterine cervical cancer develops mainly due to persistent infection with certain types of human papillomavirus (HPV). Progression to invasive cervical cancer is slow and infrequent.

In Japan, funding for HPV vaccination for girls aged 12–16 years began in 2010. However, serious adverse events after HPV vaccination were widely reported in the Japanese media [1–4]. Repeated news reports on the occurrence of diverse symptoms, including chronic pain, motor impairment, and other symptoms in some vaccine recipients arose. Therefore, the Japanese Ministry of Health, Labour and Welfare (MHLW) announced the suspension of the governmental recommendation of HPV vaccination in 2013. When the suspension was first announced, the MHLW announced that it would continue until accurate information could be made available to the public; however, the suspension has been continuing for more than 5 years. These events combined to negatively affect Japanese mothers' intention to vaccinate their adolescent daughters. Many of them have seemed to assume that HPV vaccine is a toxic substance that has a negative effect on their adolescent daughters' nerves. The inoculation rate

**Funding:** The authors received no specific funding for this work.

**Competing interests:** The authors declare that no competing interests exist.

has sharply declined. Vaccine coverage subsequently dropped to less than 1% and has remained this low to date (= the HPV vaccine crisis in Japan).

Although the benefits of HPV vaccination in teens with regard to cancer prevention have been reported to outweigh the risks and potential side effects related to vaccine administration [5–7], the actual situation in Japan is that the outcomes of examinations on the influence of HPV vaccine are contradictory [8,9]. Based on analyses using the same data from the Nagoya City Surveillance Survey, Yaju and Tsubaki [8] examined a possible association between HPV vaccination and distinct symptoms such as cognitive impairment or movement disorders, while Suzuki and Hosono [9] reported that HPV vaccinations are not significantly correlated with the occurrence of serious symptoms. The longer the uncertainty around Japan's HPV vaccine suspension, the more public concerns will grow [7]. In Japan, for example, the World Health Organization (WHO)'s Global Advisory Committee on Vaccine Safety has commented that young women are vulnerable to preventable HPV-related cancers, and that the policy decisions in Japan are resulting in the lack of HPV vaccination and inability to decrease cervical cancers in Japan [10].

If HPV vaccination is not recommended, a strategy to prevent cervical cancer through effective screening may be essential, and all efforts to increase examination rates should be continued. However, such screening-based expectations have not been met, especially in young women. Based on the National Life Basic Survey conducted by the Japanese MHLW [11], the consultation rate for uterine cancer screening is low, at about 40%, in asymptomatic women in Japan, despite it being subsidized by public funds. On the other hand, the consultation rate on prenatal visits is high, at about 99% [12]. Based on these backgrounds, in Japan uterine cancer screening during pregnancy is now also subsidized by public funds. In the Guidelines for Obstetrical Practice in Japan 2017 edition [13], screening for cancer of the uterine cervix using a cytological examination is now highly recommended for women during an early stage of pregnancy.

Cervical screening is based on taking a sample of superficial cervical cells for the detection of atypical cells associated with malignant transformation. In the Guidelines for Office Gynecology in Japan 2017 edition [14], as the appropriate way of obtaining samples for cervical cytology, spatula or brush (including 'broom' types) use is strongly recommended for non-pregnant women, while a cotton swab is allowed to collect cell samples from pregnant women because the uterine cervix during pregnancy is fragile and hemorrhages easily [15,16]. When the results of a smear indicate atypical squamous cells of undetermined significance (ASC-US), high-risk HPV testing, colposcopy, or repeat cytology conducted after 6 months is recommended [6,14]. Colposcopy is recommended in cases of positive testing results for high-risk HPV. Pregnancy has been suggested to influence the false-positive rate of malignant cytology based on a previous observation using liquid-based cytology (LBC) and conventional cervical cytology (Pap test) [17]; however, LBC has been reported to be more accurate than conventional cervical cytology and has the potential to optimize the effectiveness of primary cervical cancer screening [18,19].

Based on these backgrounds, we examined the current status of the results of cervical cytology conducted during pregnancy in Japan. In this study, we also examined the appropriate sampling/cytology methods for cervical cytology during pregnancy.

## Materials and methods

The protocol for this study was approved by the Ethics Committee of the Japan Association of Obstetricians and Gynecologists (JAOG). Because no individual can be identified under the protocol of this retrospective study of medical records, the ethics committee waived the

requirement for informed consent from each subject. In addition, we confirmed that all data were fully anonymized before analyzing them.

In April 2019, we requested 2,293 obstetrical facilities that are JAOG members to provide information on screening for cancer of the uterine cervix using a cytological examination subsidized by public funds in pregnant women who delivered at $\geq$ 22 weeks' gestation between October 1, 2018 and March 31, 2019. A total of 1,292 (55.5%) of the 2,330 obstetrical facilities responded with valid information on a total of 238,743 women, accounting for approximately 51% of all deliveries that occurred in Japan during the study period (approximately 460,000 births in 6 months).

In the current study, inquiries other than those about the prevalence of abnormal cervical cytology, other than being negative for an intraepithelial lesion or malignancy (NILM) in the Bethesda system, were as follows: (1) maternal age at delivery, (2) spatula/brush or cotton swab as sampling methods, (3) LBC or conventional cervical cytology, and (4) additional tests and the results of ASC-US.

The $X^2$ or Fisher's exact test was used for categorical variables. Differences with $p < 0.05$ were considered significant.

## Results

Of the 1,262 institutions that responded with valid information, 810 (64.2%) used a cotton swab while 842 (66.7%) used conventional cervical cytology.

The implementation rate of cervical cytology during pregnancy subsidized by public funds was 86.8% in Japan (**Table 1**). There were no significant differences in the rate between age groups.

The prevalence of abnormal cervical cytology by maternal age was 3.3% (**Table 2**). ASC-US and a low-grade squamous intraepithelial lesion (LSIL) accounted for 59.1 (3,973/6,727) and 25.4% (1,712/6,727), respectively. The prevalence of ASC-US and LSIL in teenagers was significantly higher than that in those of other ages.

The prevalence of high-risk HPV positive women with ASC-US by maternal age was 65.3% (**Table 3**). Although the prevalence of positive high-risk HPV in teenagers was significantly higher than women of other ages ($p < 0.01$), there were no significant differences in the rate between age groups.

The prevalence of abnormal cervical cytology during pregnancy using a spatula/brush with LBC was 4.9% (**Table 4**). The detection rate of abnormal cervical cytology with LBC was higher than that with conventional cervical cytology, regardless of sampling methods ($p < 0.01$). In addition, in cases with conventional cervical cytology, the detection rate of abnormal cervical cytology using a spatula/brush was higher than the one with a cotton swab ($p < 0.01$).

## Discussion

The main findings of this study show the high prevalence of abnormal cervical cytology as well as high-risk HPV in pregnant teenagers. In addition, the overall prevalence of abnormal

**Table 1. Implementation rate of uterine cervical cytology during pregnancy in women by maternal age.**

| Maternal age (y) | Total number | Number of cytology examination | Examination rate (%) |
|---|---|---|---|
| -19 | 2,550 | 2,285 | 89.4 |
| 20–29 | 82,655 | 72,435 | 87.6 |
| 30–39 | 139,569 | 119,909 | 85.9 |
| 40- | 13,961 | 12,130 | 86.9 |
| Total | 238,735 | 206,759 | 86.6 |

**Table 2. Prevalence of abnormal uterine cervical cytology by maternal age.**

| Maternal age (y) | Total | Abnormal uterine cervical cytology | | | | |
|---|---|---|---|---|---|---|
| | | Total | ASC-US | LSIL | HSIL | SCC |
| -19 | 2,285 | 150 (6.6) | 87 (3.8) | 51 (2.2) | 9 (0.4) | 0 (0) |
| 20–29 | 72,435 | 2,674 (3.7)* | 1,590 (2.2)* | 741 (1.0)* | 292 (0.4) | 5 (0.0) |
| 30–39 | 119,909 | 3,467 (2.9)* | 2,044 (1.7)* | 818 (0.6)* | 479 (0.4) | 14 (0.0) |
| 40- | 12,130 | 436 (3.6)* | 252 (2.1)* | 102 (0.8)* | 61 (0.5) | 1 (0.0) |
| Total | 206,759 | 6,727 (3.3) | 3,973 (1.9) | 1,712 (0.8) | 841 (0.4) | 20 (0.0) |

Data are presented as number (percentage).

ASC-US, atypical squamous cells of undetermined significance.

LSIL, low-grade squamous intraepithelial lesion.

HSIL, high-grade squamous intraepithelial lesion.

SCC, squamous cell carcinoma.

*$P < 0.01$ vs. women aged $\leq$ 19 years.

cervical cytology during pregnancy was 3.3% by conventional cervical cytology (Pap testing) and 4.9% using LBC.

## Prevalence of abnormal cervical cytology in women by age

According to previous studies [20,21], an abnormal cervical cytology is more frequent in pregnant women compared with the general population, and the majority is ASC-US although pregnant women with abnormal cervical cytology tend to show regression after delivery, possibly due to shedding of cervical epithelial cells during delivery. The higher prevalence of a HPV-positive result in ASC-US groups has been reported to be associated with a younger age [34]. These findings are in line with similar studies with non-pregnant study, in which the positive rate of HPV ranged from 30–50% [22,23]. The current results support these previous observations [20–24]. It could be related to the risk factor of an early age of the first intercourse. Because women with high-risk HPV-positive results were actually infected with HPV before pregnancy, the results can be extrapolated to the general population of women in Japan. In addition, these prevalence rates in Japan do not differ from those reported from other countries such as Sweden and Germany [25,26].

The current prevalence rate of abnormal cervical cytology in pregnant women with the highest rates among teenagers is comparable with our previous observation regarding

**Table 3. Implementation rate of high-risk HPV test and the prevalence of positive high-risk HPV in women with ASC-US by maternal age.**

| Maternal age (y) | ASC-US | high-risk HPV test | high-risk HPV-positive |
|---|---|---|---|
| -19 | 87/2,285 (3.8) | 47/87 (54.0) | 33/47 (70.2) |
| 20–29 | 1,590/72,435 (2.2) | 1,033/1,590 (65.0) | 596/1,033 (57.7)* |
| 30–39 | 2,044/119,909 (1.7) | 1,334/2,044 (65.3) | 621/1,334 (46.6)* |
| 40- | 252/12,130 (2.1) | 179/252 (71.0) | 56/179 (31.3)* |
| Total | 3,973/206,759 (1.9) | 2,593/3,973 (65.3) | 1,306/2,593 (50.4) |

Data are presented as number (percentage).

HPV, human papillomavirus.

ASC-US, atypical squamous cells of undetermined significance.

*$P < 0.01$ vs. women aged $\leq$ 19 years.

**Table 4. Prevalence of abnormal uterine cervical cytology by sampling/cytology methods and maternal age.**

| Sampling methods | Spatula/brush | | Cotton swab | |
|---|---|---|---|---|
| Cytology methods | conventional cervical cytology | LBC | conventional cervical cytology | LBC |
| Maternal age (y) | | | | |
| -19 | 16/345 (4.6) | 40/480 (8.3) | 64/1,133 (5.6) | 30/327 (9.2) |
| 20–29 | 385/9,897 (3.9) | 802/14,219 (5.6) | 937/38,310 (2.4) | 550/10,009 (5.5) |
| 30–39 | 535/16,487 (3.2) | 923/21,736 (4.2) | 1,374/65,824 (2.1) | 635/15,862 (4.0) |
| 40- | 68/1,666 (4.1) | 101/2,003 (5.0) | 197/6,954 (2.8) | 70/1,507 (4.6) |
| Total | 1,004/28,395 (3.5)* | 1,866/38,438 (4.9)*# | 2,572/112,221 (2.3) # | 1,285/27,705 (4.6)*# |

Data are presented as number (percentage).

LBC, liquid-based cytology.

* $P < 0.01$ vs. those by conventional cervical cytology with cotton swab.

# $P < 0.01$ vs. those by conventional cervical cytology with sptula/brush.

Condylomata acuminate (CA), which is one of the most common sexually transmitted diseases caused by HPV infection [27,28]. In previous studies, which reported age-based estimates, younger participants had higher HPV or CA prevalence estimates than older participants associated with cervical epithelial cell-related immaturity and their sexual behavior because the numbers of sexual partners are the highest in these younger age groups [29]. In an earlier study in Japan [30], for example, the prevalences of HPV infection in women aged 15–19, 20–24, 25–29, and ≥ 30 years were 44, 29, 20, and 7%, respectively. Therefore, younger women tend to exhibit a higher prevalence of HPV infection. Considering the relatively low prevalence (about 50%) of HPV testing in teenagers diagnosed with ASC-US, it may be necessary to re-encourage HPV vaccination for teenagers. Because when screening tests are applied to many subjects, a high ratio of false-positive results may be disconcerting, and false-positive results in screening may lead to unnecessary biopsies and treatments [28]. Otherwise, a more frequent and regular cervical examination for teenagers who chose to forego HPV vaccination will be required, although cervical cancer screening is now subsidized for women over the age of 20 in most parts of Japan [11]. We now fear that suspension of the HPV vaccine recommendation may further decrease the rate of screening young Japanese women compared with other developed countries [31].

The HPV infection rate was observed to fall markedly at the age range from 30 to 35 years old, which may well correspond with the Japanese social phenomenon whereby the average age of marriage for a woman is over 30 years [30]. Therefore, it is feared that HPV infection is becoming more prevalent in young couples those have become more active in sexual intercourse with the background of the HPV vaccine crisis in Japan [1–4].

Therefore, the HPV vaccine crisis in Japan may disturb the decreasing rate of abnormal cytology observed in other countries.

## Prevalence of abnormal cervical cytology by sampling/cytology methods

For cervical cytology, important elements to consider are collection procedure, specimen storage and sample preparation [28]. Although a large variety of methods for taking cervical swabs is available, further development of devices for the self-collection of vaginal samples is ongoing in the world. However, there have seemed to be no clear methods for cervical cytology in Japan.

In this study, there were significant differences in the prevalence rate of abnormal cervical cytology between sampling/cytology methods. The detection rate of abnormal cervical

cytology with LBC was higher than that with conventional cervical cytology, regardless of sampling methods. In addition, in cases with conventional cervical cytology, the detection rate using a spatula/brush was higher than that using a cotton swab. The current results support the clinical usefulness of spatula/brush for gynecological cytology during pregnancy [17–19,32,33]. However, in this study the high detection rate of abnormal cervical cytology using a cotton swab with LBC (4.6%). To our knowledge, there have been no established studies for examining the accuracy of cervical cytology using a cotton swab with LBC, except for one report in Japan [29]. In the study [29], a clear background was noted in 90% of the samples using a cotton swab with LBC; however, the detection rate of abnormal cervical cells seemed to be low (4%). This may not be sufficient examination to advocate the usefulness of cotton swabs for cervical cytology, as cytologic evaluation has not been performed.

In this retrospective study, the data cannot reflect the detection rate because of the two different methods for different samples. However, we can say that a higher prevalence of abnormal cervical cytology was noted in the LBC group. Indeed, a previous study [34] used both methods in a single case and compared detection rates, and revealed that the rate of unsatisfactory cytology was lower with LBS but there was no significant difference in the detection of epithelial cell abnormalities. We also understand that there are some additional limitations, such as a bias in the details of cell sampling even with the same methods. Differences between various spatulas/brushes or LBC were also not examined. In addition, differences in gestational weeks when cancer screening was performed were not considered. Because the evaluation of cytologic screening during pregnancy has been reported to be likely to be underestimated itself [35,36], a further study that unifies these conditions may be needed to improve the accuracy of cervical cancer screening during pregnancy.

## Conclusion

Prevention of cervical cancer constitutes a public health priority, and vaccine introduction should be programmatically feasible even in Japan; however, unfortunately there have been no clear cervical cancer prevention programs in Japan. Because greater benefit and protection from the vaccine is thought to come from immunising preadolescent individuals [37–39], increasing vaccination coverage among teenages in Japan should still be a more cost-effective primary objective. In addition, based on the current results cervix cytodiagnosis has been performed by non-uniform methods in Japan. Because a high prevalence of abnormal cervical cytology was noted in teenagers, building systems and methods to effectively conduct cervical cytology examination will be needed if they choose to forego HPV vaccination.

## Acknowledgments

We thank JAOG members for their cooperation with our questionnaire. We thank the Japanese Foundation for Sexual Health Medicine for their excellent suggestions.

## Author Contributions

**Conceptualization:** Shunji Suzuki.

**Data curation:** Shunji Suzuki, Eijiro Hayata, Shin-ichi Hoshi.

**Formal analysis:** Shunji Suzuki.

**Investigation:** Shunji Suzuki.

**Methodology:** Shunji Suzuki, Eijiro Hayata, Shin-ichi Hoshi, Akihiko Sekizawa, Yoko Sagara.

**Project administration:** Shunji Suzuki, Akihiko Sekizawa, Yoko Sagara, Masanobu Tanaka, Katsuyuki Kinoshita, Tadaichi Kitamura.

**Resources:** Shunji Suzuki.

**Software:** Shunji Suzuki.

**Supervision:** Shunji Suzuki.

**Validation:** Shunji Suzuki.

**Visualization:** Shunji Suzuki.

**Writing – original draft:** Shunji Suzuki, Eijiro Hayata.

**Writing – review & editing:** Shunji Suzuki, Shin-ichi Hoshi, Akihiko Sekizawa, Yoko Sagara, Masanobu Tanaka, Katsuyuki Kinoshita, Tadaichi Kitamura.

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
