## [Decision Letter · Decision Letter 0]

26 Nov 2020

PONE-D-20-27341

Current status of uterine cervical cytology during pregnancy in Japan

PLOS ONE

Dear Dr. Suzuki,

Thank you for submitting your manuscript to PLOS ONE. After careful consideration, we feel that it has merit but does not fully meet PLOS ONE’s publication criteria as it currently stands. Therefore, we invite you to submit a revised version of the manuscript that addresses the points raised during the review process.

We look forward to receiving your revised manuscript.

Kind regards,

Magdalena Grce, PhD

Academic Editor

PLOS ONE

Journal Requirements:

2. Please provide additional details regarding participant consent.

In the ethics statement in the Methods and online submission information, please ensure that you have specified (i) whether consent was informed and (ii) what type you obtained (for instance, written or verbal, and if verbal, how it was documented and witnessed).

If your study included minors, state whether you obtained consent from parents or guardians. If the need for consent was waived by the ethics committee, please include this information.

3. Thank you for stating the following financial disclosure: 'No'

4. Thank you for stating the following in your Competing Interests section: 'No'

a. Please complete your Competing Interests statement to state any Competing Interests. If you have no competing interests, please state "The authors have declared that no competing interests exist.", as detailed online in our guide for authors at http://journals.plos.org/plosone/s/submit-now

Reviewers' comments:

Reviewer's Responses to Questions

**Comments to the Author**

1. Is the manuscript technically sound, and do the data support the conclusions?

Reviewer #1: Yes

Reviewer #2: Partly

2. Has the statistical analysis been performed appropriately and rigorously? 

Reviewer #1: Yes

Reviewer #2: N/A

3. Have the authors made all data underlying the findings in their manuscript fully available?

Reviewer #1: Yes

Reviewer #2: Yes

4. Is the manuscript presented in an intelligible fashion and written in standard English?

Reviewer #1: Yes

Reviewer #2: No

5. Review Comments to the Author

Reviewer #1: The manuscript has a good potential. However, the impression is that the potential is not realized and exploited enough, which leaves a reader with the overall pale notion, almost as if the manuscript is somehow incomplete or truncated.

The high potential lies in the large cohort that is analyzed (includes 51% of all pregnant women in Japan appeared in a consecutive period of six months or approximately quarter of million women). This holds large scientific relevance per se, irrespective of the results.

The results are far from pale and their implications need to be further and more thoroughly addressed in the discussion part of the manuscript, which is its weakest segment.

Here are some specific suggestions for further improvement :

1. The Discussion section of the manuscript:

The backbone of the manuscript results that is most interesting and stands out is present in the Table 3. It shows that abnormal cervical test results ranked in the category of the lowest degree of abnormality (Atypical squamous cells of undetermined significance /ASC-US/) are strongly associated with the high-risk HPV strain viral infection only in (pregnant) teenage women (70% of women), while the same relation in all other, elder / non-teenage (pregnant) women is bellow 50% on average.

Considering the initial large cohort and the reasonable possibility that the high-risk HPV positive pregnant women are actually infected before pregnancy, the results can, in rough, be extrapolated on a general population of women in Japan.

Taking this into account it would be interesting for the authors to address the following:

a) What would be the authors opinion regarding current Japanese Government recommendation for suspension of the HPV vaccination for teenage women? What exact results stand behind the general perception of lack of safety for the HPV vaccine in Japan?

b) Could the alternative be the recommendation for a more frequent and regular cervix examination for the teenage women that opt to omit the HPV vaccination?

c) Considering relatively low prevalence of the HPV testing in the Japanese teenage women diagnosed with ASC-US cervical abnormality ( around 50%), could the manuscript results suggest raising the prevalence rate to almost obligatory level for HPV testing in those types of women?

2. The Introduction section:

a) It would be helpful and interesting to expand some more about the HPV vaccine crisis in Japan. For example, what are particular adverse effects of the HPV vaccine that are observed in Japan and what are their average occurrence prevalence in inoculated Japanese women? In that way, a reader could draw more clearly its own conclusion about the cost/benefit ratio on HPV vaccination.

On the Page 7, Lines 14-16, of the manuscript stands: Therefore, the HPV vaccine crisis in Japan may disturb the decreased rate of the abnormal cytology that will be observed ? in other

countries. Are there any, at least preliminary, results regarding HPV vaccination of teenage women in some other developed countries and its effect on the (ab)normality of the female population cervical cytology or You just assume that there is one? It would be effective to briefly mention some other developed countries policies regarding HPV vaccination in the Introduction section.

b)The Methods section (Page 4, Line 14 - Page 5, Line 10), which explains Guidelines for gynecology in Japan on the way of obtaining samples for cervical cytology, would be more fit to transfer and incorporate into the Introduction section.

Language:

Although the structure of sentences are simple and generally comprehensible, the manuscript needs additional language editing as there are some, more or less serious, errors that can affect the paper semantics. Some examples:

- missing the subject and the verb: Page 8 , Line 11 : «In this study, there are differences..«

- missing important adjective : Page 9, Line 6: »abnormal cervical cells«

- probable misprint: Page 9, Line 8: safely instead of »safety«

- different preposition/conjunction.: Page 3, Line 11: for instead of »from«,

Page 9 , Line 15. which instead of »with«

- different verb: Page 2, Line 20: develops instead of »is«

Reviewer #2: Overall, the study is interesting, comprehensive, and the data are valuable. It is good designed but the presentation, that is all parts of the manuscript (introduction, material and methods, results and discussion) should be improved and re-written. The whole study should be changed with the English language corrections and written in a more scientific way.

In general, the manuscript comprises too little references, too short text in general, especially discussion part, and the data are presented only in tables without diversity.

English language should be improved in a scientific way and professionally edited.

Introduction: too little literature, scarce data, and mainly about funding; lack the data about cervical cancer, HPV, vaccine, ... The sentence: ”However, the serious adverse events after HPV vaccination were widely reported in the Japanese media.” (page 3, line 14-15) is reported in the introduction part, but without more detailed explanation and/or references.

Materials and methods are only one paragraph; it should be separated on the study group section and the sections of used methods. Some parts are not suitable for this chapter (page 5, line 6-10). There is not study group characterized as it is, nor methods descripted or assigned literature, for example using terms “as described previously…”

Results chapter: lack the introduction part. Results should be more comprehensive and precisely written. Data from the tables should be pointed in the text as well. In Table 1, and in the main text, the term “performing rate of uterine cervical cytology” is not clear enough terminology.

Discussion is not well written; lack the introduction and overall data presentation in the first part, as well as the literature data and references. The second part is better written (page 8, line 18-23). The end of discussion, the conclusion part should be improved; for example, it can’t start with “We understand that there are some other limitations such as a bias in the details of cell sampling even with the same instrument.” (page 9, line 9-10).

6. PLOS authors have the option to publish the peer review history of their article (what does this mean?). If published, this will include your full peer review and any attached files.

Reviewer #1: No

Reviewer #2: No

---

## [Author Response · Author response to Decision Letter 0]

8 Dec 2020

December 7, 2020

Editorial office 

PLOS ONE

Dear Editors, 

We would like to thank you and the reviewer for the comments and critique of my manuscript entitled ‘Current status of uterine cervical cytology during pregnancy in Japan’. We have been able to respond positively to each comment and we believe the paper has been strengthened. The changes are highlighted as red colored text.

The protocol for this study was approved by the Ethics Committee of the Japan Association of Obstetricians and Gynecologists (JAOG). Because no individual can be identified under the protocol of this retrospective study of medical records, the ethics committee waived the requirement for informed consent with each subject. In addition, we have confirmed that all data were fully anonymized before we accessed them.

The authors received no specific funding for this work.

Therefore,

1. We have re-confirmed PLOS ONE style.

2. We have added the comments concerning consent in the Methods.

3. There are no financial disclosure.

4. There are no COI.

In addition, the data are published in Figshare (10.6084/m9.figshare.13347299, Title: Current status of cervical cytology in Japan).

Responses to Reviewer 1,

Many thanks for your careful reading of the manuscript. We appreciate your comments very much. Thank you very much for your suggestions. We have re-written the manuscript heavily, relying on your suggestions.

1. Thank you very much for your suggestion in the Discussion. With your suggestion, we realized the need for a detailed examination of the results of teenage women. We have added the comments concerning the Japanese Government recommendation (a), examination methods (b) and examination rate of HPV testing for teenage women (c).

2. In the Introduction, we have re-written to add the comments of HPV vaccine crisis in Japan. In addition, we have changed the Introduction and Methods as suggested.

3. Thank you very much for your corrections. The manuscript has been re-checked by an English native speaker.

Responses to Reviewer 2,

Many thanks for your careful reading of the manuscript. We appreciate your comments very much. Thank you very much for your suggestions. We have re-written the manuscript heavily, relying on your suggestions.

1. We have added some references to add the comments of vaccine crisis in Japan.

2. The manuscript has been re-checked by an English native speaker.

3. We have changed the Introduction and Methods as suggested.

4. We have added the introduction and comments of tables.

5. We have separated the Discussion to 2 parts. We have added the Conclusion.

We do hope and trust that with these changes the manuscript is now acceptable for publication.

Thank you very much, again.

Sincerely yours,

Shunji Suzuki, MD 

Department of Obstetrics and Gynecology, 

Japanese Red Cross Katsushika Maternity Hospital 

5-11-12-2 Tateishi, Katsushika-ku, Tokyo 124-0012 Japan 

Tel: +81-3-3693-5211 

Fax: +81-3-3694-8725 

e-mail: czg83542@mopera.ne.jp

---

## [Editor Report · Decision Letter 1]

17 Dec 2020

PONE-D-20-27341R1

Current status of uterine cervical cytology during pregnancy in Japan

PLOS ONE

Dear Dr. Suzuki,

Thank you for submitting your manuscript to PLOS ONE. After careful consideration, we feel that it has merit but does not fully meet PLOS ONE’s publication criteria as it currently stands. Therefore, we invite you to submit a revised version of the manuscript that addresses the points raised during the review process.

Before resubmission your manuscript should be edited by a professional.

We look forward to receiving your revised manuscript.

Kind regards,

Magdalena Grce, PhD

Academic Editor

PLOS ONE

Additional Editor Comments (if provided):

General changes to do:

CCC is not a usual abbreviation for conventional cervical cytology, therefore please replace it with the name as it is or with Pap (Papanikolaou) smear / Pap test / Pap testing wherever it is necessary (manuscript and tables).

Do not use the term incidence as you are not evaluating it in this study. The term prevalence is appropriate.

Uterine cervix cytology is equivalent to cervical cytology, so use the term “cervical cytology”.

Define a teenage age range in a Materiel and Method section and replace “teenage women” by “teenagers”.

Results/Abstract:

The expression “than that” is incorrect. So, in the Abstract, page 8 lines 10 and 15, page 9 line 5 and 7, page 12 line 11, please replace:

- „than that in those of other ages“ by „than women of other ages“ or „than older women than xx years;

- “with LBC was higher than that with CCC” by “with LBC was higher than for Pap testing”

- „than that using a cotton swab“ by „than the one with a cotton swab“

Avoid starting a sentence with “Table X shows” and emphasise the subject. Consider the following construction:

- The implementation rate of uterine cervical cytology during pregnancy subsidized by public funds was 86.8% in Japan (Table 1).

- The prevalence of abnormal uterine cervical cytology by maternal age was 3.3% (Table 2). – delete the next sentence.

- The prevalence of high-risk HPV positive women with ASC-US by maternal age was 65.3% (Table 3). – delete the 1st sentence.

- The prevalence of abnormal uterine cervical cytology during pregnancy using a spatula/brush with LBC was 4.9% (Table 4). – delete the 1st sentence.

Discussion/Abstract:

Please correct the 1st sentence into:

The main findings of this study shows the high prevalence of abnormal cervical cytology as well as high-risk HPV in pregnant teenage women. In addition, the overall prevalence of abnormal cervical cytology during pregnancy was 3.3% by Pap testing and 4.9% using LBC.

Page 9, line 15: correct construction of the phrase would be: “Prevalence of abnormal cervical cytology in women by age” OR simply “Age prevalence of abnormal cervical cytology”

Page 10, line 1: delete some

Page 10, line 6: provide a reference

Page 10, line 7: “The findings paralleled those of a non-pregnant study” replace with “These findings are in line with similar studies with non-pregnant women”

Page 10, line 9: “an early age at the” replace with “an early age of the”

Page 10, line 9: complete the sentence “other countries such as …… and …… [24,25].”

Page 10, line 15-17: “our previous observation in Condylomata acuminate (CA), which is one of the sexually transmitted diseases caused by HPV infection.” replace with “our previous observation regarding Condylomata acuminate (CA), which is one of the most common sexually transmitted diseases caused by HPV infection”; see IARC Monographs on the Evaluation of Carcinogenic Risks to Humans: Human papillomaviruses, 2007

Page 11, line 1-2: “cervical biology immunity”, did you mean with “cervical epithelial cell-mediated immunity”?

Page 11, line 7: “(= around 50%)” replace by “(about 50%)”

Page 11, line 8-9: the assumption and suggestion is wrong; see the recommendation of IARC (IARC Handbooks of Cancer Prevention: Cervix Cancer Screening, 2005)

Page 11, line 14: provide a reference for HPV vaccine recommendation in Japan or at least in the world.

Page 11, line 15-16: “from around 30-35 years old” replace by “at the age range from 30 to 35 years”

Page 11, line 17: add women, “the average age of marriage for a woman is over 30 years”

Page 11, line 17-18, page 12, lines 1-2: the sentence is unclear and should be rewritten. What do you mean by Westernized lifestyle and HPV vaccine crisis?

Page 12, lines 3-4: same observation as for the previous sentence. What do you mean by HPV vaccine crisis in Japan? Please, provide exact facts and references for your statements.

Page 12, line 12: consider “methods” instead of “instruments”

Pages 12-14 on the evaluation of sampling and testing methods should be completely revised taking into consideration the sensitivity and the specificity of each methods in your study compared to the published studies.

Page 14, Conclusion should be reconsidered regarding the changes that has to be done to this study/manuscript. If there are no clear cervical cancer prevention programs (primary program, HPV vaccination and secondary program, cervical screening) in Japan please consider carefully 1) the IARC recommendation on cervical cancer prevention as a golden standard, and 2) the recommendation of World Health Organization on HPV Immunization as well as the European Centre for Disease Prevention and Control (ECDC) guidance on the introduction of HPV vaccines in European countries (2012) also as a golden standard, before making any conclusions.

---

## [Author Response · Author response to Decision Letter 1]

22 Dec 2020

Dear Editors, 

We would like to thank you and the reviewer for the comments and critique of our manuscript entitled ‘Current status of (uterine) cervical cytology during pregnancy in Japan’. We have been able to respond positively to each comment and we believe the paper has been strengthened. The changes are highlighted as red colored text.

The protocol for this study was approved by the Ethics Committee of the Japan Association of Obstetricians and Gynecologists (JAOG). Because no individual can be identified under the protocol of this retrospective study of medical records, the ethics committee waived the requirement for informed consent with each subject. In addition, we have confirmed that all data were fully anonymized before we accessed them.

The authors received no specific funding for this work.

Therefore,

1. We have re-confirmed PLOS ONE style.

2. We have added the comments concerning consent in the Methods.

3. There are no financial disclosure.

4. There are no COI.

In addition, the data are published in Figshare (10.6084/m9.figshare.13347299, Title: Current status of cervical cytology in Japan).

Responses to the Editor

Many thanks for your careful reading of the manuscript. We appreciate your comments very much. Thank you very much for your suggestions. We have re-written the manuscript heavily, relying on your suggestions.

We have changed from CCC to conventional cervical cytology. Otherwise, we have used the word of Pap smear as appropriate. We have changed to ‘prevalence’, ‘cervical cytology, and ‘teenagers’.

Results/Abstract: Thank you very much for your suggestions. We have corrected the sentences/words according to your suggestions.

Discussion/Abstract: We have corrected the sentences/words according to your suggestions. We have re-learned the recommendations from European specialists based on the literatures you suggested (Ref: 28,37,38). We have reflected on our own statements in the first revised manuscript. We have re-written the Discussion & Conclusions with the European recommendations. In addition, we have explained the HPV vaccine crisis in the Introduction

Responses to Reviewer 1,

Many thanks for your careful reading of the manuscript. We appreciate your comments very much. Thank you very much for your suggestions. We have re-written the manuscript heavily, relying on your suggestions.

1. Thank you very much for your suggestion in the Discussion. With your suggestion, we realized the need for a detailed examination of the results of teenage women. We have added the comments concerning the Japanese Government recommendation (a), examination methods (b) and examination rate of HPV testing for teenage women (c).

2. In the Introduction, we have re-written to add the comments of HPV vaccine crisis in Japan. In addition, we have changed the Introduction and Methods as suggested.

3. Thank you very much for your corrections. The manuscript has been re-checked by an English native speaker.

Responses to Reviewer 2,

Many thanks for your careful reading of the manuscript. We appreciate your comments very much. Thank you very much for your suggestions. We have re-written the manuscript heavily, relying on your suggestions.

1. We have added some references to add the comments of vaccine crisis in Japan.

2. The manuscript has been re-checked by an English native speaker.

3. We have changed the Introduction and Methods as suggested.

4. We have added the introduction and comments of tables.

5. We have separated the Discussion to 2 parts. We have added the Conclusion.

We do hope and trust that with these changes the manuscript is now acceptable for publication.

Thank you very much, again.

Sincerely yours,

Shunji Suzuki, MD 

Department of Obstetrics and Gynecology, 

Japanese Red Cross Katsushika Maternity Hospital 

5-11-12-2 Tateishi, Katsushika-ku, Tokyo 124-0012 Japan 

Tel: +81-3-3693-5211 

Fax: +81-3-3694-8725 

e-mail: czg83542@mopera.ne.jp

---

## [Editor Report · Decision Letter 2]

26 Dec 2020

Current status of uterine cervical cytology during pregnancy in Japan

PONE-D-20-27341R2

Dear Dr. Suzuki,

We’re pleased to inform you that your manuscript has been judged scientifically suitable for publication and will be formally accepted for publication once it meets all outstanding technical requirements.

Kind regards,

Magdalena Grce, PhD

Academic Editor

PLOS ONE
---

## [Editor Report · Acceptance letter]

30 Dec 2020

PONE-D-20-27341R2 

Current status of cervical cytology during pregnancy in Japan 

Dear Dr. Suzuki:

I'm pleased to inform you that your manuscript has been deemed suitable for publication in PLOS ONE. Congratulations! Your manuscript is now with our production department. 

Kind regards, 

on behalf of

Dr. Magdalena Grce 

Academic Editor

PLOS ONE